# A Novel Swept-Back Fishnet-Embedded Microchannel Topology

**DOI:** 10.3390/mi14091705

**Published:** 2023-08-31

**Authors:** Yan Wang, Xiaoyue Zhang, Xing Yang, Zhiji Wang, Yuefei Yan, Biao Du, Jiliang Zhang, Congsi Wang

**Affiliations:** 1School of Information and Control Engineering, Xi’an University of Architecture and Technology, Xi’an 710055, China; ywang@xauat.edu.cn (Y.W.); wangzhiji0512@163.com (Z.W.); 2School of Mechano-Electronic Engineering, Xidian University, Xi’an 710071, China; yaungx@foxmail.com; 3Guangzhou Institute of Technology, Xidian University, Guangzhou 510555, China; congsiwang@163.com; 4CETC No.54 Research Institute, Shijiazhuang 050081, China; biaodu@163.com (B.D.); zhangjiliang_work@163.com (J.Z.)

**Keywords:** embedded cooling, microchannel, heat dissipation, heat exchange

## Abstract

High in reliability, multi in function, and strong in tracking and detecting, active phased array antennas have been widely applied in radar systems. Heat dissipation is a major technological barrier preventing the realization of next-generation high-performance phased array antennas. As a result of the advancement of miniaturization and the integration of microelectronics technology, the study and development of embedded direct cooling or heat dissipation has significantly enhanced the heat dissipation effect. In this paper, a novel swept-back fishnet-embedded microchannel topology (SBFEMCT) is designed, and various microchannel models with different fishnet runner mesh density ratios and different fishnet runner layers are established to characterize the chip Tmax, runner Pmax, and Vmax and analyze the thermal effect of SBFEMCT under these two operating conditions. The Pmax is reduced to 72.37% and 57.12% of the original at mesh density ratios of 0.5, 0.25, and 0.125, respectively. The maximum temperature reduction figures are average with little change in maximum velocity and a small increase in maximum pressure drop across the number of fishnet runner layers from 0 to 4. This paper provides a study of the latest embedded thermal dissipation from the dimension of a single chip to provide a certain degree of new ideas and references for solving the thermal technology bottleneck of next-generation high-performance phased array antennas.

## 1. Introduction

Active Phased Array Technology enables radars to have high performance and high survivability while reducing the cost of radar development, so active phased array radar antennas are widely used in satellite imaging, aircraft early warning, battlefield reconnaissance, ground air defence, and other fields [1]. Active phased array antennas are composed of many active components and electronic devices. Because of the miniaturization of microelectronics technology, the development trend of high integration—making microelectronics devices work when the heat flux per unit area within the device increased significantly—is currently the main reason for the performance of the electronic devices to fail with declining appreciation. According to statistics, the reliability of the system will decrease by 50% when the temperature of the electronic devices increases by 10 °C. The cooling system can reduce electronic device temperature and maintain electrical performance stability, which has become an indispensable part of electronic device design [2,3,4]. The proposal and research of embedded cooling, i.e., direct cooling, has shifted the direction of chip cooling from traditional remote cooling to proximity cooling, i.e., the cooling mass is delivered directly into the chip substrate or adapter plate for cooling. This approach to heat dissipation ignores the numerous interface materials and component housings that impede heat dissipation, thus minimizing the total thermal resistance between the chip junction and the terminal thermal sink and achieving efficient heat dissipation [5]. In addition, the embedded heat sink uses a vertical flow, which shows a lower flow MLD relative to the horizontal flow. This is attributed to the effect of jet impingement at the manifold surface that prevented the appearance of flow recirculation, which contributed to a better hydrothermal performance than the horizontal flow [6]. This new approach to thermal control opens more possibilities for today’s thermal dilemmas.

Microchannels, which are small channels etched or fabricated on a substrate, are one of the existing emerging technologies for electronics cooling. These emerging techniques for electronics cooling include heat pipes, microchannels, spray cooling, phase change material (PCM)-based cooling, thermoelectric cooling, free cooling, liquid immersion cooling, and 3D stacked architectures with interlayer microchannel cooling. These emerging techniques for electronics cooling offer innovative solutions to address the challenges posed by the use of microchannels, and by high heat-generating electronic devices. Microchannels work by providing a large surface area for heat exchange to enhance heat transfer [7]. Microchannel cooling technologies can be divided into two categories: active and passive. The difference between active and passive technologies is that active technologies require an external power source to drive them while passive technologies do not. Active techniques include electrostatic force, pulsed flow and vibration; passive techniques include design improvements and working fluid improvements [8]. Currently, in the direction of working fluid improvement, nanofluids are considered to have great potential for future cooling technologies. Nanofluids can enhance thermal conductivity, improve convective heat transfer, and enhance boiling heat transfer. Furthermore, nanofluids have compatibility with existing cooling systems and the potential to revolutionize cooling technologies for electronics [7]. Of course, there are many challenges accompanying their application, such as the use of carbon-based nanoparticles in nanofluids; more consideration should be given to production and environmental protection when using carbon-based nanoparticles [9].

In addition, incompressible thermo-fluids are also widely used. Incompressible thermo-fluid refers to a fluid that is considered to be incompressible and exhibits thermodynamic properties. Incompressible fluids are those in which the density remains constant regardless of changes in pressure. This assumption is valid for many practical engineering applications involving liquids, such as water or oil, where the density changes are negligible. However, some important difficulties were previously encountered in solving incompressible flows using pressure correction schemes, and artificial compressibility methods were developed. The artificial compressibility method helps in solving the continuity equation by introducing an artificial compressibility term into the equations. This term allows the continuity and momentum equations to be coupled, enabling the use of time-marching methods. By adding artificial compressibility, the convergence process can be improved without affecting the results. The study [10] proposed a multidimensional artificial characteristic-based (MACB) scheme, which has various applications for solving incompressible flows with heat transfer accurately, like in heat exchangers, vortex tubes and complex geometries.

Embedded cooling is described in the previous work [11] as a third-generation thermal management technology for electronic circuits and is a research focus in DARPA’s Near-Junction Thermal Transport (NJTT) and Intra/Inter-Chip Enhanced Cooling Thermal Packaging (ICECool) programs. Li Zheng et al. [12] proposed a novel stacked embedded microfluidic cooling system for chip I/O interconnects, enabling high broadband signals, embedded microfluidic cooling, and power transfer for high-performance stacked ICs. An embedded cooling system was demonstrated by John Ditr [13] and others. The coolant crosses the interposer material directly into the interior of the chip package, closer to the actual heat-generating transistors in the chip than the coolant of a cold-plate-based cooling solution. This cooling solution reduces the total thermal resistance by a factor of three or more compared to the total thermal resistance of conventional remote cooling methods due to the many reduced thermal interfaces. The thermal performance of the designed embedded cooling system is also experimentally verified.

Remco van Erp [14,15] proposed a new thermal management method for liquid-cooled GaN-based power integrated circuits embedded directly in Si substrates, with PCBs as the delivery platform. The performance coefficient of a manifold thermal structure etched into the Si substrate is increased by a factor of 50 over that of a straight microchannel. This co-design of microfluidic channels and electronics in semiconductor substrates can further improve integration. Furthermore, pure silicon surfaces are typically hydrophobic. Hydrophobic surfaces are known for their self-cleaning properties, low water absorption capabilities, and resistance to wetting. Also, hydrophobic surfaces reduce the frictional resistance between the fluid and the channel walls, resulting in lower pressure gradients and less resistance to flow. The slip velocity of the fluid near the hydrophobic surface increases, promoting better mixing and heat transfer between the fluid and the wall. This results in improved heat transfer performance in microchannels with hydrophobic surfaces [16]. In the previous work [17], microchannel heat sinks were integrated into Si substrates for GaN-based power integrated circuits. This solution reduces the thermal resistance by a factor of 25 compared to forced-air cooling. Experimental measurements show that integrated liquid cooling reduces the impact of self-heating on electrical performance. Research teams such as Lockheed Martin and Purdue University have conducted studies and experiments related to embedded cooling, and the results show that the direct cooling of the fluid embedded in the chip has a higher thermal efficiency than the traditional convective heat transfer method based on liquid-cooled cold plates. As the latest and most advanced cooling method, many countries have studied and transitioned to embedded cooling, and the experimental data they have compiled have confirmed the superior cooling capability of embedded cooling. In China, however, there is still a certain technical shortcoming in microelectronics technology for the time being, and there is very little research work and related reported studies on embedded cooling. Siyuan Miao [18] introduced the latest and most advanced cooling method—embedded cooling for high-power phased array antenna chips—and conducted a simulation analysis. Chaotin Li [5] designed two associated cooling schemes for a single chip, the swept-back breakpoint microchannel and the truncated microchannel, with the truncated microchannel cooling scheme being an enhancement of the swept-back breakpoint microchannel cooling scheme. The optimized truncated microchannel shows a 0.81% reduction in maximum temperature and an 8.4% increase in temperature uniformity compared to the through-microchannel cooling structure. This indicates that increasing the width of the spoiler fins is beneficial in enhancing thermal performance. The truncated microchannel structure is a flat microchannel in the first half and a breakpoint + fin flow channel in the second half. According to the results, it can be seen that this flow channel can effectively enhance the thermal performance, but at the same time the maximum pressure drop increases significantly and the heat mostly collects in the chip wafer in the middle and lower part of the flow channel, and since the breakpoint + fin flow channel is distributed horizontally and perpendicular to the flow direction of the coolant flowing through the flat channel, there are many areas in the fin runners where effective contact is not possible.

Therefore, this paper proposes a novel swept-back fishnet-embedded microchannel topology (SBFEMCT), which differs from the truncated microchannel in that the rectangular break point and fins are changed into a diamond lattice to increase the effective contact area and to a certain extent strengthen the disturbance structure in the middle of the microchannel to disturb the fluid boundary layer and weaken the disturbance structure on both sides of the flow channel. This further improves the heat transfer performance and effectively lowers the maximum pressure drop. In this paper, the latest embedded heat dissipation technology is investigated from the dimension of a single chip to provide new ideas and references to solve the next generation of high-performance phased array antenna heat dissipation technology bottlenecks.

## 2. Design Methods/Models

### 2.1. Design Ideas

This design has been conceived because the highest temperature of the chip wafer when the chip is operating is on the central axis, and the high-temperature range in the centre is larger than on the sides, while the temperature on the sides of the chip wafer is slightly lower [5]. Therefore, to better dissipate heat through the liquid coolant in the embedded flow channel, it can be heavily weighted to bring out the heat generated in the mid-axis, the central part of the chip, during operation. To achieve heat dissipation enhancement, structures are often designed to block or direct the fluid to change its original flow direction when designing heat dissipation structures.

For example, this can be achieved by adding a pin-fin structure at the bottom of the runner or adding a flow disturbance structure at the side to disturb the fluid boundary layer [19,20], or preparing porous micro-runners, etc. The previous work [21] showed that fin geometries can be varied, and certain fin geometries, such as staggered or oblique configurations, can induce secondary flow passages and promote better mixing and turbulence. This can result in improved heat transfer by disrupting the thermal boundary layer and enhancing convective heat transfer. In conclusion, the micro pin fin configuration offers several advantages compared to the conventional microchannel shown in the previous work [22]: Firstly, the micro pin fin configuration provides a larger surface area for heat transfer, leading to improved heat dissipation. Secondly, the presence of pin fins promotes a secondary (lateral) flow, which enhances fluid mixing and heat transfer. Thirdly, the flow around the pin fins creates vortices, which further enhance heat transfer by disrupting the thermal and hydraulic boundary layers. Fourthly, the pin fins facilitate better mixing of the coolant, leading to improved heat distribution and reduced temperature non-uniformity. In addition, fins with larger surface areas can provide more contact between the coolant and the heated surface, leading to enhanced heat transfer [21,23].

The study [24] investigated reinforced structures such as oblique secondary channels and rectangular ribs. The results show that a heat dissipation structure with projecting rectangular ribs + inclined secondary channels shows a significant improvement in maximum temperature and homogeneity compared to the flat microchannel structure. In the previous work [5], a flow channel with a flat microchannel in the first half and a breakpoint + fin in the second half was designed. According to the results, this six-channel can effectively enhance the thermal performance, but at the same time, the maximum pressure drop increases significantly. In addition, the breakpoint + fin runner is distributed horizontally, perpendicular to the flow of the coolant through the flat channel. This results in many areas of the breakpoint + fin flow path not being effectively contacted and the heat dissipation effect needing to be improved. The number of prism pin fin sides has a significant impact on the performance of the heat sink, which was proposed in the previous work [25]. The study found that the four-sided configuration of the micro prism pin fin heat sink had the best performance. The four-sided configuration exhibited prominent secondary flow, which led to efficient fluid mixing and enhanced heat transfer. Therefore, the SBFEMCT in this paper uses flat microchannels in the first half of the runner and near the outlet, with a diamond lattice design in the middle and rear sections. The breakpoints in the section near the central axis are extended towards the entrance of the runner, and the breakpoints in the longitudinal direction near the edge are shortened towards the exit. The overall triangular shape of the rhombic flow channel section and the overall symmetry of the flow channel are along the longitudinal central axis. The lattice not only forces the coolant in the microchannels to be disturbed and mixed laterally between adjacent microchannels but also increases the effective contact area. The overall triangular shape of the rhombic lattice increases the lateral and diagonal perturbation, which better disperses the heat on the central axis to the sides and rear. This design enhances thermal performance without severely reducing the flow rate of the fluid while minimizing the cost of the simulation.

### 2.2. Designed Model

SBFEMCT is shown in Figure 1, with the arrow pointing to the embedded microchannel, which is embedded in the substrate beneath the chip. The size of the heat source chip is set to 3.5 mm × 2.2 mm × 0.2 mm. The operating temperature range is known to be 0 °C–70 °C for commercial chips and −40 °C to −85 °C for industrial chips. The heat flux of the chip is set to 0.5 [W]/Volume(chip)[cm^3^], which is approximately 330 [W/cm^3^] in the experiment. The heat flux density can be calculated as about 50 [W/cm^2^], and the maximum wafer temperature is obtained as 51.67 °C, which both meet the operating temperature requirements of the chip [26].

When a fishing net is pulled up from the sea, the grid is mostly diamond-shaped, and all the water in the net flows downward quickly along the sides of the diamond grid, because of the constant diversion and convergence of the mesh structure. Inspired by this, the microchannel is designed as a fishing net structure. In this paper, silicon is chosen as the structural material, water is used as the coolant, the height of the microchannel is fixed at 600 µm, and in order to improve the heat dissipation performance as much as possible, the width of the mesh-like flow channel is taken as 100 µm. The middle and back part of the microchannel is composed of the mesh-like diamond lattices, and the rest of the part consists of the flat microchannel connecting the inlet and outlet of the microchannel. The width of the flat microchannel at the entrance is determined by the diamond lattice on the diagonal side of the mesh flow channel closest to the entrance, and its width is equal to the distance from the node of the previous diamond lattice near the side to the node of the next diamond lattice near the side. The width of the flat microchannels at the outlet is determined by the other row of the mesh flow channel near the outlet and is equal to the width of the central axis of the rhombus. The middle 3 flat microchannels at the outlet and the 2 flat microchannels near the edge are suitably widened to reduce the pressure drop and increase the outflow rate. The flat microchannels at the inlet are aligned with the corners of the diamond lattice so that the cooling mass can be diverted whenever it enters the mesh microchannels, due to the bifurcation structure formed by the diamond lattice. After diversion, the cooling medium converges with the cooling medium already inside the mesh microchannel, and then diverges again due to the bifurcation structure. In addition, when the cooling medium is in the mesh microchannel, secondary fluid flow occurs when it hits the wall. The bifurcation phenomenon and the secondary fluid flow in the model of this paper are shown in Figure 2. Constant diversion and convergence can improve the effective contact area and force the coolant in the microchannels to enable lateral perturbation and doping between neighbouring microchannels, which leads to enhanced heat dissipation performance.

As the flow channel is symmetrical, the mesh density ratio of the mesh flow channel is defined as the inverse of the number of lattices in the top left-hand diagonal row of the mesh flow channel. For example, in the model in Figure 1, there are eight lattices in the top left-hand diagonal row of the flow channel, so the mesh density ratio of the flow channel is 1/8 = 0.125.

Due to the high alignment of the rhombic lattice, the mesh flow channel has a high degree of ductility throughout the flow channel. Specifically, the mesh can be extended or contracted in the longitudinal direction with the same number of longitudinal straight runners, grid density ratio, and mesh width. Thus, the number of fishnet runner layers is defined in this paper as the number of rhombuses beyond the fourth straight runner from the centre to the left. As in the model in Figure 2, the number of rhombuses in the lower left part of the flow channel beyond the fourth longitudinal straight flow channel from the centre to the left is 1, so the number of layers in the flow channel is 1.

## 3. Simulation Experiments

This study uses the computational fluid software ANSYS workbench 2022 R1 to solve the flow and heat transfer problems of SBFEMCT based on a 3D coupled model. ANSYS Workbench is a co-simulation environment proposed by ANSYS, a new front-end interface alongside ANSYS Classic (Mechanical APDL). Furthermore, the software ANSYS is publicly recognised in the industry and can be used to calculate with relatively accurate results [27,28]. It allows the analysis and simulation of structural statics, structural dynamics, rigid body dynamics, fluid dynamics, structural thermodynamics, electromagnetic fields, and coupled fields of complex mechanical systems. The finite element analysis process for the workbench is divided into three steps: “pre-processing + analysis + post-processing”. Each of these steps is divided into three sub-steps: pre-processing—model construction + material definition and assignment + meshing; solution—load boundary + displacement boundary conditions + solution setup; and post-processing—conventional results (stress, strain, deformation) + path + export of results.

In this study, the 3D model is first modelled using Design Modeler on the workbench and the fluid domain is obtained using the Fill function. The model then meshes in workbench meshing. The meshed model is then imported into Fluent for setup and solution. Before initializing the flow field and solving the calculations, the following simplifications and assumptions are made: the flow field is unidirectional, laminar, and incompressible; the physical properties of the fluid and solid heat sink are constant; the fluid flows at a constant rate with no slip against the fixed walls; and the solid–liquid interface meets the conditions of temperature uniformity, heat flow continuity, and no sliding boundary layers [2,5,29,30]. When using Fluent, the mesh quality of the model is first checked to ensure that the minimum volume of the mesh is greater than zero. If this condition is not met then we return to meshing for mesh refinement. After this, the solver is set up and the basic equations to be solved are selected. For this study, the pressure solver is chosen and all simulations are set to steady-state analysis. As the model analysis involves coupled heat-flow analysis, the energy equations need to be opened. The Reynolds number calculation determines that the fluid flow state for this study is laminar, which means that laminar flow needs to be selected. Next, materials need to be added and boundary conditions need to be given and set. The same materials were used for all the simulations in this study and the materials and their parameters were set as shown in Table 1. In the boundary condition settings for the experiments, the inlet is set to velocity inlet, the inlet velocity is set to 1.0 m/s by default, and the inlet temperature is set to 20 °C. The outlet is the pressure outlet with a relative static pressure of 0. The wall is set to adiabatic conditions, i.e., no consideration is given to radiative heat transfer and convective heat transfer with air. For continuity, it is considered that the velocity calculation results converge to a residual of less than 10-6 and the energy equation has a residual of less than 10-9. Therefore, the continuity equation can be expressed as below.
(1)∇⋅u=0

The momentum equation can be expressed as the following form.
(2)$$\left(u\cdot \nabla \right)\rho_{f}u=-\nabla p+\mu \nabla^{2}u$$

The energy equation for the fluid domain and for the solid domain can be expressed as (3) and (4), respectively.
(3)$$\rho_{f}c_{p,f}\left(u\cdot \nabla T\right)=k_{f}\nabla^{2}T$$
(4)$$k_{s}\nabla^{2}T=0$$

### 3.1. Feasibility of SBFEMCT

#### 3.1.1. A Runnerless Model

The appropriate heat flux value of the heat source is determined by a simulation experiment in a runnerless model. The size of the heat source chip is set to 3.5 mm × 2.2 mm × 0.2 mm. The operating temperature range is known to be 0 °C–70 °C for commercial chips and −40 °C to –85 °C for industrial chips. The heat flux of the chip is set to 0.5 [W]/Volume(chip)[cm^3^], which is approximately 330 [W/cm^3^] in the experiment. In the post-processor CFD-Post the heat source, i.e., the heat-generating chip, is selected individually to display its temperature cloud, and the maximum wafer temperature is obtained as 51.67 °C, which meets the operating temperature requirements of the chip.

#### 3.1.2. SBFEMCT

The SBFEMCT is constructed and its heat dissipation effect is analyzed. The flow rate is 1.0 m/s and the chip size and its heat flux values are the same as in the no-flow channel model. The model is shown in Figure 2. Enter the Mesh module for meshing. The mesh is encrypted and the resulting finite element model is shown in Figure 3, containing a total of 8,001,646 mesh nodes and 17,413,469 mesh cells.

The solved temperature field of the chip is shown in Figure 4, with a maximum chip temperature of 21.05 °C. The maximum temperature of the chip is 21.05 °C. The maximum temperature of the chip is 69.3% lower for the swept-back mesh-embedded microchannel topology compared to the no-runner condition. The pressure and velocity fields of the flow path are shown in Figure 5, which shows a pressure drop of 3231 Pa in the chip cooling flow path of SBFEMCT.

### 3.2. Analysis of the Effect of Flow Rate on the Heat Dissipation Effect of SBFEMCT

The coolant used in the model was aqueous, and the thermal performance of the model was analyzed by varying the inlet velocity value of the coolant. After meshing, the finite element model was obtained and Fluent was started according to the previous procedure, setting the heat flux of the chip to that of the no-fluid model, approximately 330 [W/cm^3^]. The pressure outlet was set to a relative static pressure of 0. The surfaces of both the substrate and the solder layer were set to adiabatic wall surfaces. The inlet temperature was set to 20 and the inlet flow rate was set from 0.2 m/s to 2.6 m/s to analyze the effect of flow rate on the heat dissipation effect of the chip embedded. The maximum chip temperature, maximum air pressure in the flow channel, and maximum flow rate at different flow rates were obtained as shown in Table 2, and Figure 6 shows the maximum chip temperature at different flow rates.

As the table shows, the maximum temperature of the chip gradually decreases as the velocity increases, and the maximum atmospheric pressure and maximum flow rate in the flow channel also increase. The temperature drops rapidly with velocity in the early stages, and then the maximum chip temperature tends to level off when the velocity is greater than 1.8 m/s. This shows that the model can achieve a certain cooling effect by changing the flow rate of the coolant when the velocity is small, and the effect of speed on the cooling effect of the chip is not significant when the flow rate is large.

### 3.3. Analysis of the Effect of the Mesh Density Ratio on the Heat Dissipation Effect of SBFEMCT

As the flow channel is symmetrical, the mesh density ratio of the mesh flow channel is defined as the inverse of the number of lattices in the top left-hand diagonal row of the mesh flow channel. For example, in the model in Figure 4, there are eight lattices in the top left-hand diagonal row of the flow channel, so the mesh density ratio of the flow channel is 1/8 = 0.125.

To study the effect of the mesh density ratio on the heat dissipation effect of the chip inside the mesh flow channel part of the model, four sets of flow channels with different mesh density ratios were designed without major changes to the overall shape of the mesh flow channel and with the same width of the mesh flow channel. Namely, a mesh size of 0.9 × 0.9 with a density of 0.5, a mesh size of 0.4 × 0.4 with a density of 0.25, a mesh size of 0.15 × 0.15 with a density of 0.125, as shown in Figure 7. The flow channel with a mesh size of 0.15 × 0.15 and a density of 0.125 is the model shown in Figure 2, and its results are shown in Figure 4 and Figure 5. The results of the models with mesh size densities of 0.5 and 0.25 are shown in Figure 8 and Figure 9. The flow and heat transfer characteristics were simulated at an inlet velocity of 1.0 and the parameters obtained are shown in Table 3. The maximum pressure drop and maximum velocity of the flow channel were chosen as the characterization objects, as shown in Figure 10 and Figure 11.

It can be seen from Figure 10 and Figure 11 that the smaller the mesh density ratio, the lower the maximum pressure drop and the higher the maximum flow rate in the flow channel, but the change in mesh density ratio has a negligible effect on the maximum chip temperature. The Pmax for a grid density ratio of 0.25 is 72.37% of the Pmax for a grid density ratio of 0.5, and the Pmax for a grid density ratio of 0.125 is 57.12% of the Pmax for a grid density ratio of 0.25. The Vmax at a mesh density ratio of 0.25 is 117.3% of the Vmax at a mesh density ratio of 0.5, and the Vmax at a mesh density ratio of 0.125 is 156.4% of the Vmax at a mesh density ratio of 0.25. As the width of the fishnet runner remains the same, the fishnet runner only changes the size of the rhombus in it, so the smaller the mesh density ratio, the smaller the rhombus and the more internal runners in the fishnet runner section. Thus, the smaller the coolant area obstructed in the mesh flow channel, the lower the maximum pressure drop and the lower the maximum flow rate. More generally, refining the mesh and reducing the mesh density ratio can effectively reduce the pressure drop and flow velocity within the flow channel.

### 3.4. Analysis of the Effect of the Number of Fishnet Runner Layers on the Heat Dissipation Effect of SBFEMCT

Due to the high alignment of the rhombic lattice, the mesh has a high degree of ductility throughout the flow channel. Specifically, the mesh can be extended or contracted in the longitudinal direction with the same number of longitudinal straight runners, grid density ratio, and mesh width.

Since the runner is symmetrical from left to right as a whole, the number of fishnet runner layers is defined in this paper as the number of rhombuses beyond the fourth straight runner from the centre to the left. As in the model in Figure 2, the number of rhombuses in the lower left part of the flow channel beyond the fourth longitudinal straight flow channel from the centre to the left is 1, so the number of layers in the flow channel is 1.

To investigate the effect of the ratio of the fishnet-like flow channel compared to the overall flow channel in the model on the chip’s heat dissipation, four sets of flow channels with different numbers of fishnet-like flow channel layers were designed, which are the flow channels with 0, 1, 2, and 3 fishnet-like flow channel layers, as shown in Figure 12; its results are shown in Figure 4 and Figure 5. The flow channel with 1 fishnet-like flow channel layer is the model shown in Figure 2. The results for fishnet-like runner layers of 0, 2, and 3 are shown in Figure 13, Figure 14 and Figure 15.

In the analysis of the effect on cooling in this section, the distribution of the fishnet runners in all four models follows one principle—the upper third of the splitting point on the seam line in the fishnet runner is placed in the centre of the entire runner. This is used to ensure that the highest temperatures, in and around the centre of the chip, are disturbed more so that the temperature in this area is better brought out and a cooling effect is achieved. The flow and heat transfer characteristics were simulated at an inlet velocity of 1.0 and the parameters obtained are shown in Table 4. The highest chip temperature was chosen as the object of characterization, as shown in Figure 16.

It can be seen from Figure 16 that the higher the number of layers of the mesh flow channel, the lower the maximum temperature of the chip, the smaller the maximum pressure drop in the flow channel, and the maximum flow rate is basically unchanged. This is because the maximum temperature is too close to the inlet temperature and the maximum temperature cannot be lower than the inlet temperature, so the reduction in temperature is not obvious. The increase in the number of layers in the flow channel increases the Pmax by approximately 300 Pa. The more layers in the flow channel, the more the coolant is disturbed and mixed in the flow channel, increasing the influence on the heat exchange layer in the microchannel and thus reducing the maximum chip temperature. More generally, increasing the number of layers in the flow channel reduces the maximum chip temperature, increases the maximum pressure drop, and has little effect on the maximum flow rate.

### 3.5. Illustration of the Heat Dissipation Effect of SBFEMCT by Comparison with Traditional Microchannel Structures

Traditional microchannel structures usually follow one of two types; one is a flat microchannel type and the other is S-shaped. Therefore, the microchannel part of the SBFEMCT was changed to a flat microchannel and an S-shaped microchannel, respectively, to demonstrate the effectivity of the novel swept-back fishnet-embedded microchannel. These two models were created in Design Modeler, the same boundary conditions as SBFEMCT were set in Fluent, and then the temperature field of the whole model was obtained through thermal analysis, and the temperature results are shown in Figure 17 and Figure 18, respectively.

As shown in Figure 1 and Figure 2, the highest temperature of the flat microchannel structure type is 22.8 °C, the highest temperature of the S-type microchannel structure is 23 °C, and the highest temperature of the SBFEMCT is 21.05 °C. Through this analysis, the SBFEMCT can reduce the maximum temperature of the chip by 1.75 °C or 7.68% relative to the flat microchannel structure, while it can reduce the temperature by 1.95 °C or 8.48% relative to the S-type microchannel structure, which proves that the flow channel structure of the SBFEMCT is more effective for heat dissipation.

## 4. Conclusions

This paper proposes a novel swept-back fishnet-embedded microchannel topology (SBFEMCT) from the dimension of a single chip based on the truncated microchannel structure in the previous work [5]. The feasibility and plasticity of SBFEMCT are explored in terms of varying the fishnet-like flow channel mesh density ratio and the number of fishnet-like flow channel layers, which can be summarized as follows.

Reducing the mesh density ratio effectively reduces the pressure drop and flow velocity in the flow channel. As the mesh density ratio decreases from 0.5 to 0.25, the Pmax decreases from 7817 to 5657, the latter being 72.37% of the former; and the Vmax increases from 1.353 to 1.587, the latter being 117.0% of the former. As the mesh density ratio decreases from 0.25 to 0.125, the Pmax decreases from 5657 to 3231, the latter being 57.12% of the former; and the Vmax increases from 1.587 to 2.482, the latter being 156.4% of the former.The reduction of the grid density ratio allows a fast and effective reduction of the maximum pressure drop and a small increase of the maximum flow rate with arguably almost no effect on the maximum temperature. This can be applied to solve the problem of high pressure drop and high velocity when there is an equal requirement for maximum temperature.Increasing the number of layers of the mesh flow channel results in a general reduction in the maximum temperature, little change in the maximum velocity, and a small increase in the maximum pressure drop. The Pmax increases by approximately 300 Pa for each additional layer of the mesh flow channel.From all the temperature clouds modelled, it can clearly be seen that the highest temperature is in the chip region, which is about 21 °C, and the temperature decreases along the chip boundary in all directions, because the chip is the only heat source in the model. The fluid flows from the inlet of the microchannel, which is at an environmental temperature of 20 °C and goes underneath the chip, bringing out a certain amount of heat, thus the temperature in the area around the outlet of the microchannel is higher than that in the area around the inlet of the microchannel. In general, the temperatures are roughly symmetrical along the longitudinal centre axis.

In comparison to the model with breakpoints + fins in Section 3.1.3 of the study [5], the SBFEMCT in this paper shows a significant reduction in the maximum pressure drop, while the maximum temperature is closer to the set inlet temperature. The maximum pressure drop is one-third to one-half of that in the study [5]. Comparing the velocity cloud, it can be seen that the SBFEMCT better plays the role of shunt and convergence, and improves the effective contact area, which leads to better heat dissipation.

All the research carried out in this paper does not involve knowledge of circuits, microelectronics, etc., for the time being. Based on the success of the research in this paper, further work can be carried out on:The research in this paper is still at the simulation and analysis stage and has not yet entered the processing experiment stage. As embedded cooling is the direct pumping of the cooling mass into the chip for heat dissipation, embedded cooling has a higher heat dissipation efficiency and at the same time places more demanding requirements on the microfabrication process and packaging capabilities.The chip’s heat source is a large number of high-electron mobility transistors, which in this paper is simplified to a uniform heat-generating square. In later work, this can be modelled in detail, closer to the actual operating conditions, and its engineering significance will be greater. Accordingly, the thermal structure will need to be designed separately and may be less versatile.The thermal structures in this paper can all be further analyzed and optimized. For example, the widths of the runner entrances and exits can be refined, the widths of the fishnet-like runners can be refined, and the number of flat microchannels in the entrance and exit sections can be refined.

## Figures and Tables

**Figure 1 micromachines-14-01705-f001:**
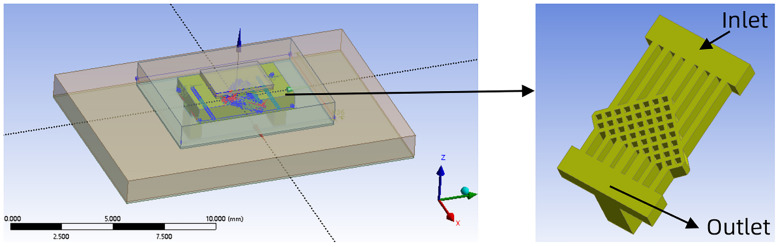
Back-swept fishnet-like embedded microchannel topology.

**Figure 2 micromachines-14-01705-f002:**
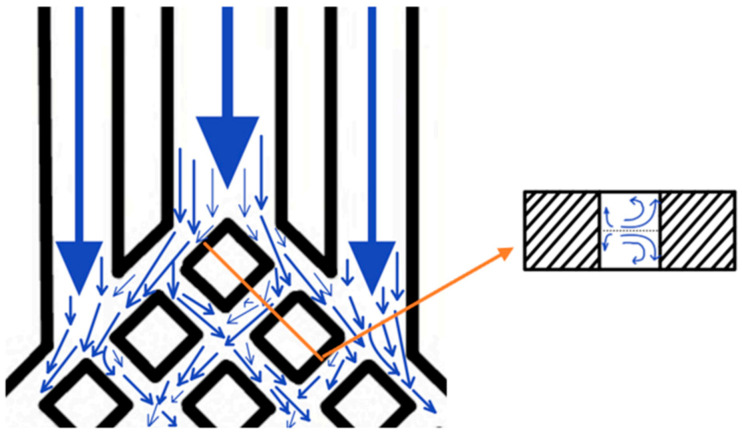
Bifurcation phenomena and secondary fluid flow.

**Figure 3 micromachines-14-01705-f003:**
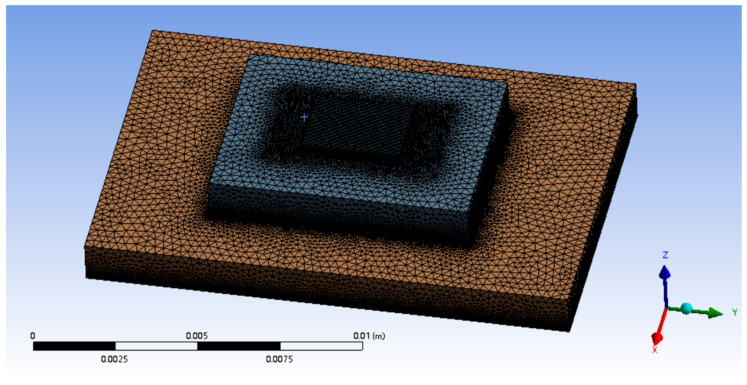
Finite element model of a Fishnet-embedded topological structure chip.

**Figure 4 micromachines-14-01705-f004:**
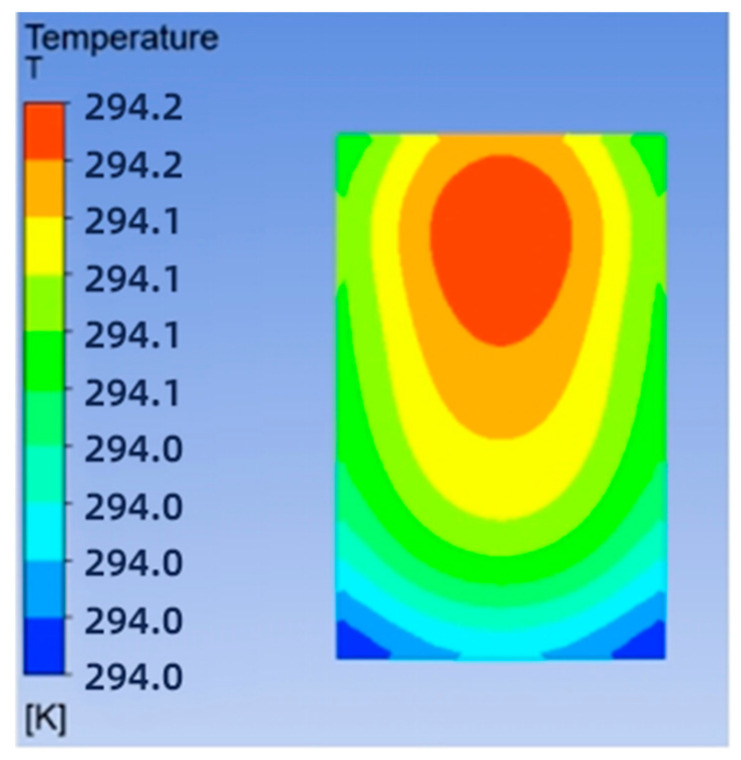
SBFEMCT chip temperature field.

**Figure 5 micromachines-14-01705-f005:**
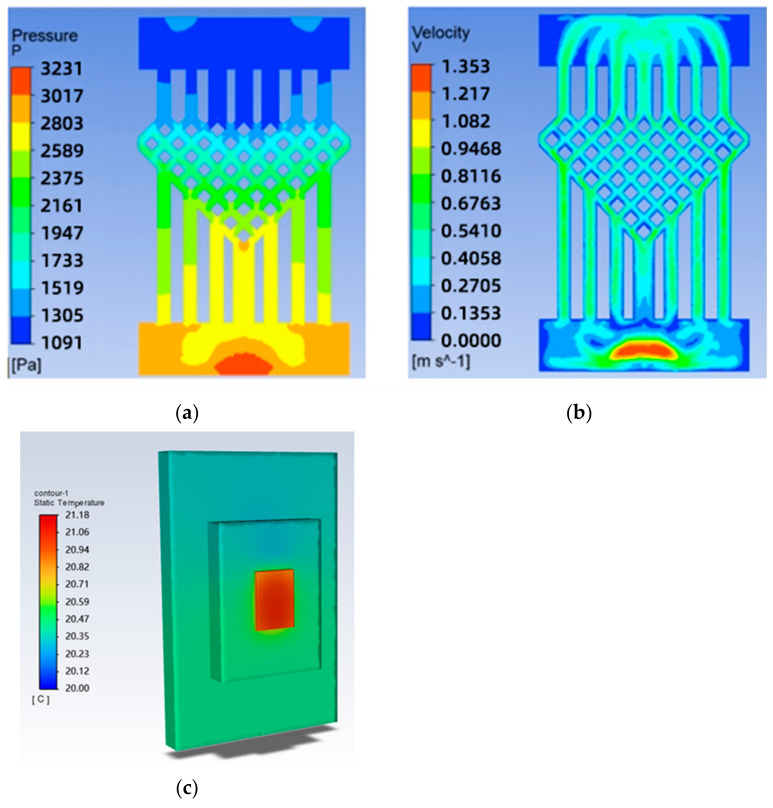
SBFEMCT flow channel pressure and velocity fields: (**a**) pressure field; (**b**) velocity field; (**c**) temperature cloud for the whole model.

**Figure 6 micromachines-14-01705-f006:**
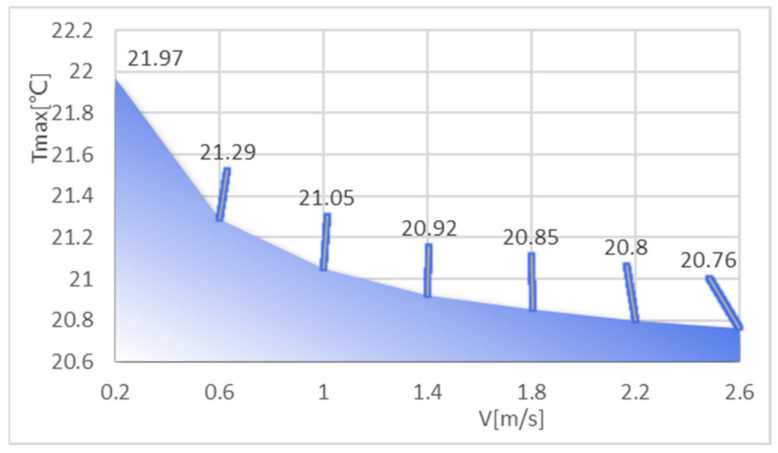
Maximum temperature of the chip at different velocities.

**Figure 7 micromachines-14-01705-f007:**
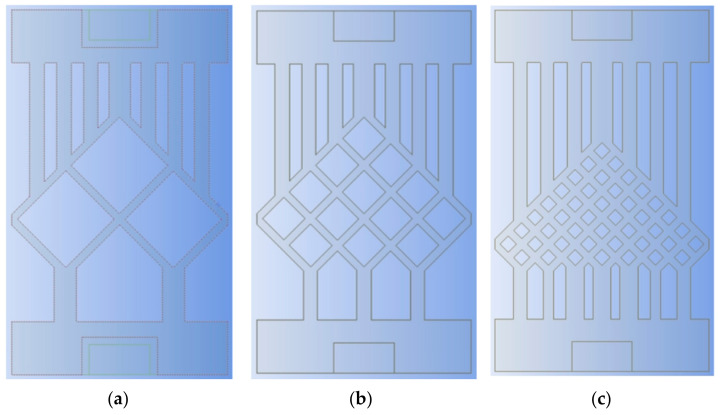
Different mesh density ratios: (**a**) The mesh density ratio 0.500; (**b**) The mesh density ratio 0.250; (**c**) The mesh density ratio 0.125.

**Figure 8 micromachines-14-01705-f008:**
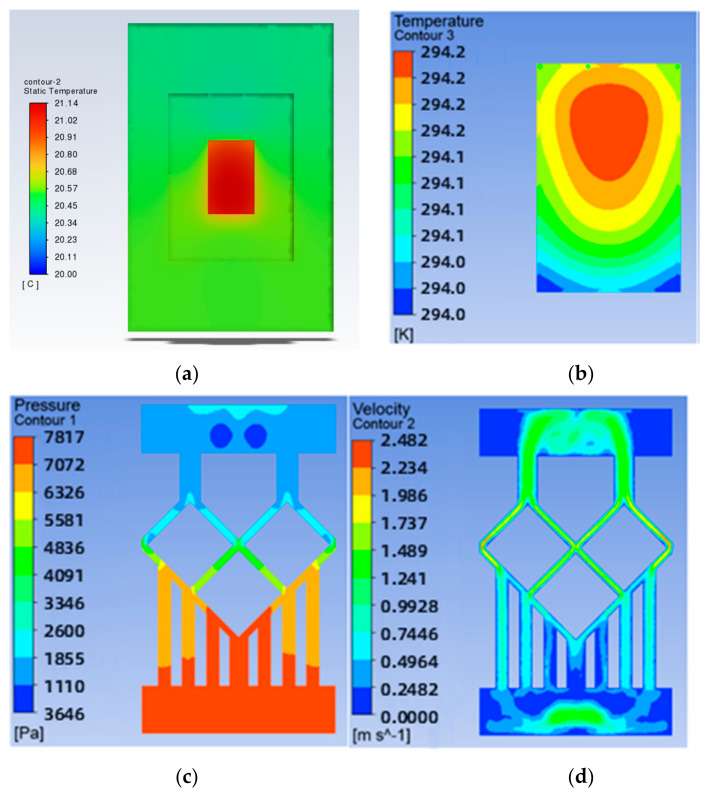
Results for a mesh density ratio of 0.500: (**a**) Temperature cloud for the whole model; (**b**) Temperature field at the chip; (**c**) Pressure field at the runner; (**d**) Velocity field at runners.

**Figure 9 micromachines-14-01705-f009:**
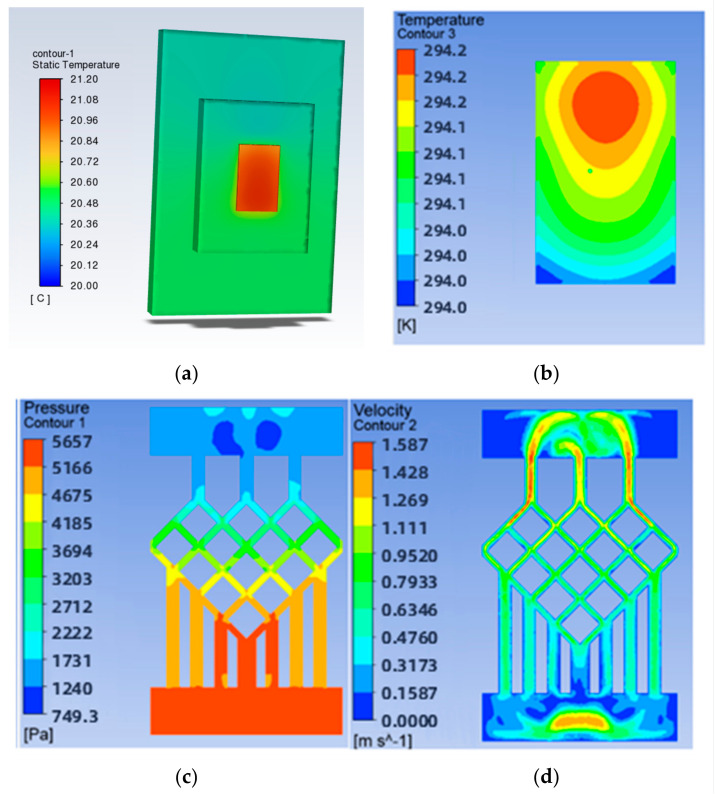
Results for a mesh density ratio of 0.250: (**a**) Temperature cloud for the whole model; (**b**) Temperature field at the chip; (**c**) Pressure field at the runner; (**d**) Velocity field at runners.

**Figure 10 micromachines-14-01705-f010:**
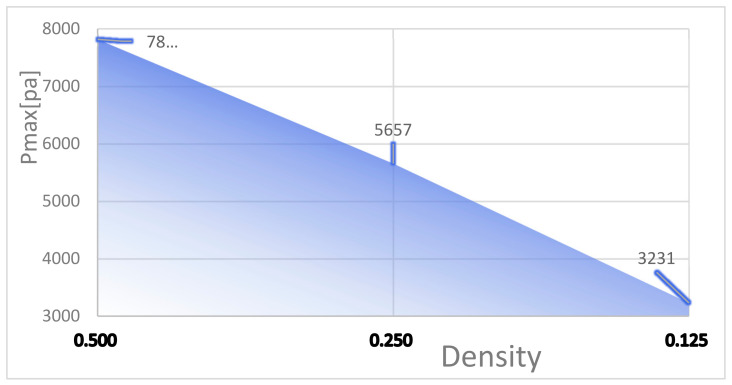
Maximum pressure drop in flow channels at different mesh densities.

**Figure 11 micromachines-14-01705-f011:**
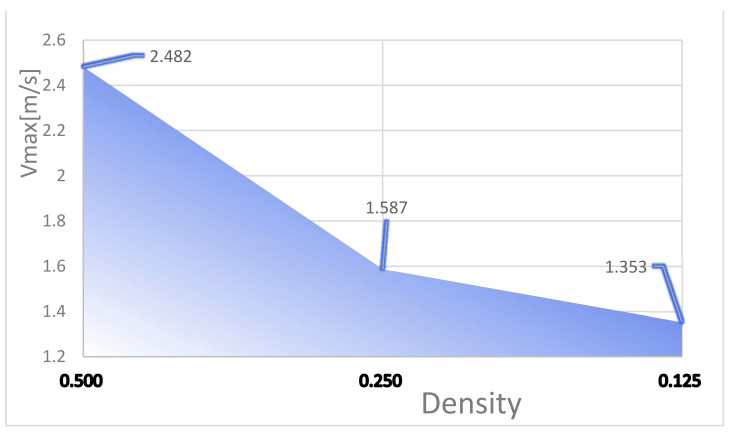
Maximum velocity in flow channels at different mesh densities.

**Figure 12 micromachines-14-01705-f012:**
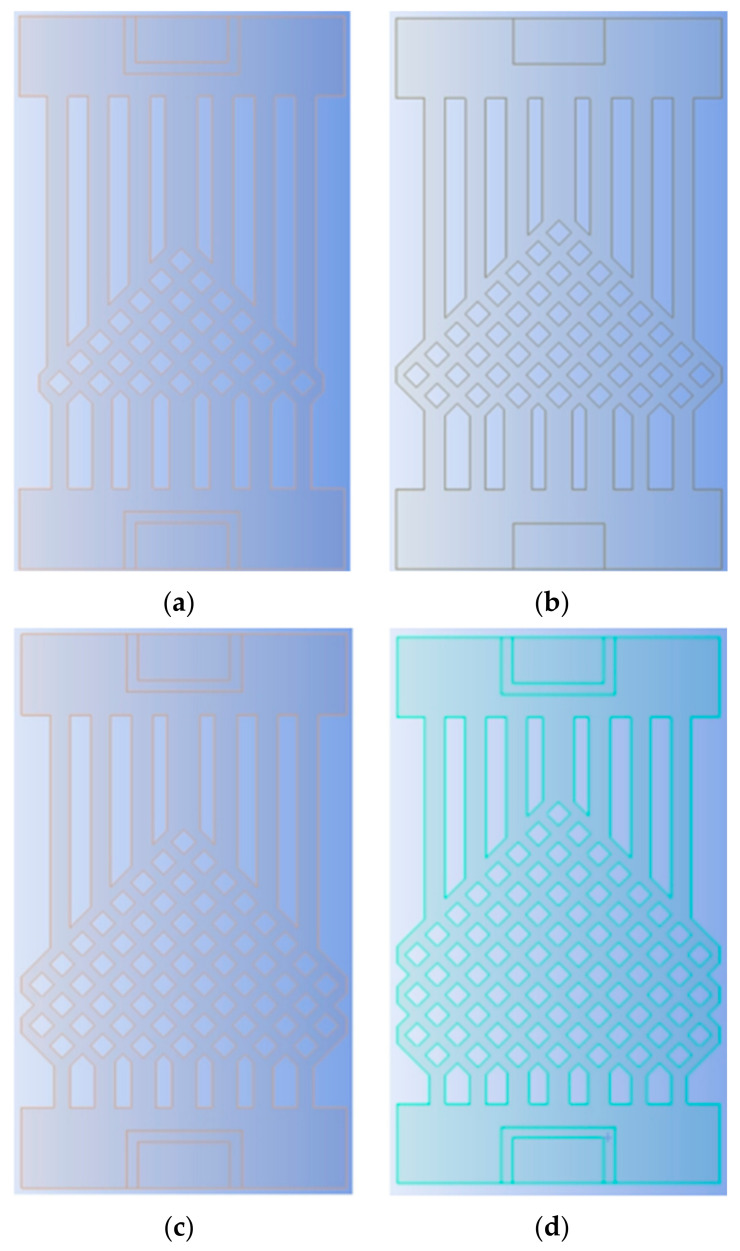
Different Fishnet Runner Layers: (**a**) 0 layers in the fishnet runner; (**b**) 1 layer in the fishnet runner; (**c**) 2 layers in the fishnet runner; (**d**) 3 layers in the fishnet runner.

**Figure 13 micromachines-14-01705-f013:**
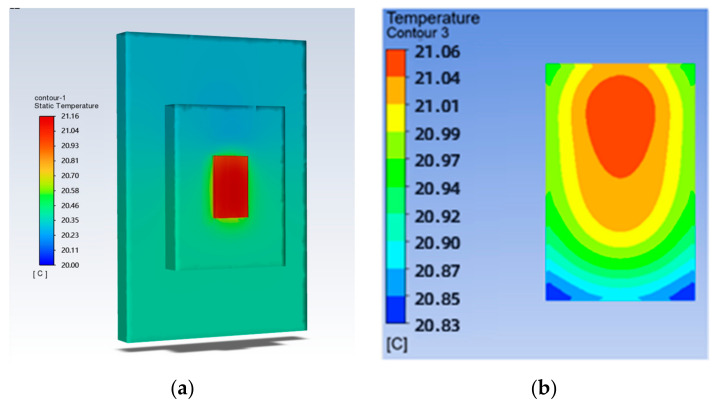
Results when the number of layers of fishnet-like runners is 0: (**a**) Temperature cloud for the whole model; (**b**) Temperature field at the chip; (**c**) Pressure field at the runner; (**d**) Velocity field at runners.

**Figure 14 micromachines-14-01705-f014:**
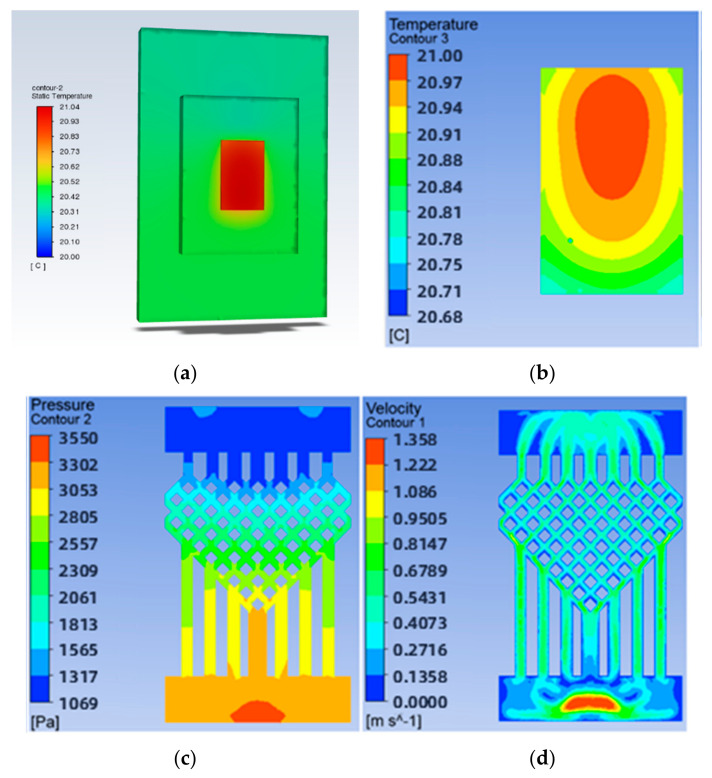
Results when the number of layers of fishnet-like runners is 2: (**a**) Temperature cloud for the whole model; (**b**) Temperature field at the chip; (**c**) Pressure field at the runner; (**d**) Velocity field at runners.

**Figure 15 micromachines-14-01705-f015:**
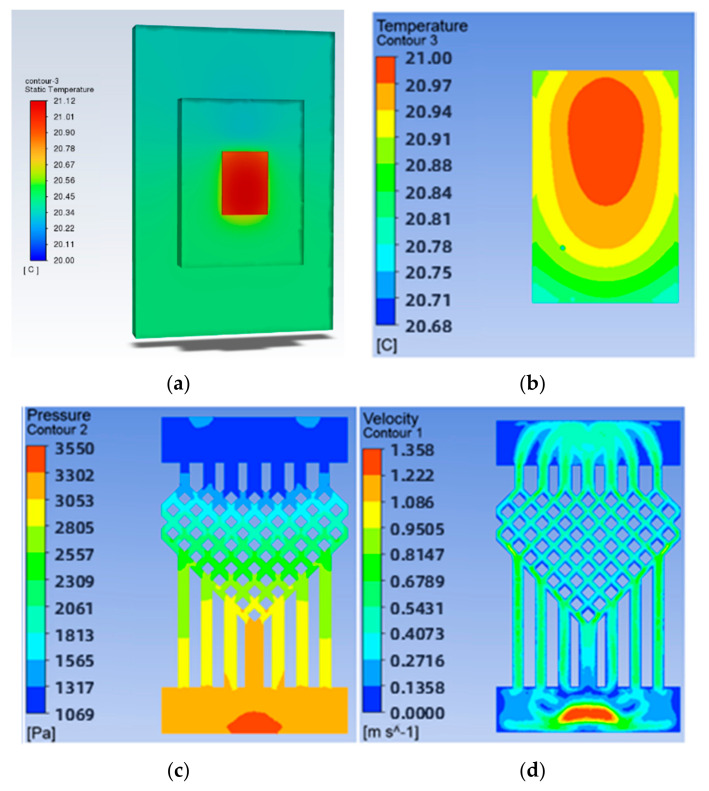
Results when the number of layers of fishnet-like runners is 3: (**a**) Temperature cloud for the whole model; (**b**) Temperature field at the chip; (**c**) Pressure field at the runner; (**d**) Velocity field at runners.

**Figure 16 micromachines-14-01705-f016:**
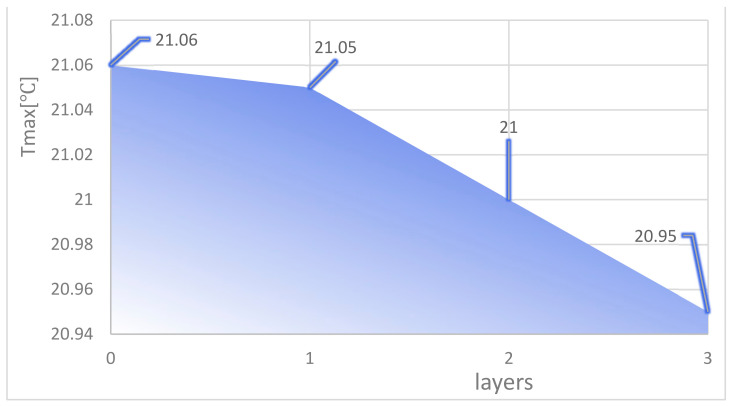
Maximum chip temperature for different layers of fishnet runners.

**Figure 17 micromachines-14-01705-f017:**
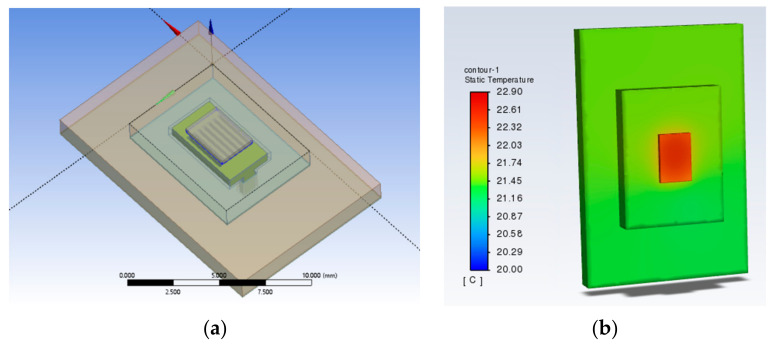
A model of flat microchannels and a temperature cloud for the whole of this model: (**a**) The model of flat microchannels; (**b**) Temperature cloud for the whole model.

**Figure 18 micromachines-14-01705-f018:**
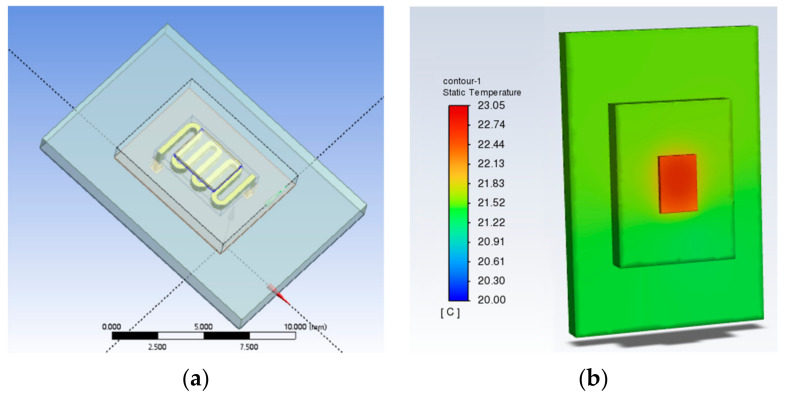
An S-shaped model and a temperature cloud for the whole of this model: (**a**) The model of flat microchannels; (**b**) Temperature cloud for the whole model.

**Table 1 micromachines-14-01705-t001:** Materials and their parameters.

Material Type	Chemical Formula	*ρ* (kg/m^3^)	*c*_p,f_ (J/(kg·°C))	Heat Conductivity [W/(m K)]	Viscosity [kg/(m s)]
solid	sn63_pb73	8400	280	10	/
	si	2300	700	150	
	htcc	3900	205	20	
	ga_as	5300	325	45	
	au_sn	14,700	150	57	
fluid	h2o <1>	998.2	4182	0.6	0.001003

**Table 2 micromachines-14-01705-t002:** Effect of different speeds on heat dissipation in SBFEMCT.

V (m/s)	Tmax (°C)	Pmax (Pa)	Vmax (m/s)
0.2	21.97	308.8	0.2971
0.6	21.29	1412	0.8491
1.0	21.05	3231	1.353
1.4	20.92	5761	1.869
1.8	20.85	8935	2.376
2.2	20.8	12,920	2.942
2.6	20.76	17,650	3.509

**Table 3 micromachines-14-01705-t003:** Heat dissipation data for different grid densities at a speed of 1.0.

Rhombus Size (mm^2^)	Density	Tmax (°C)	Pmax (Pa)	Vmax (m/s)
0.9 × 0.9	0.500	21.05	7817	2.482
0.4 × 0.4	0.250	21.05	5657	1.587
0.15 × 0.15	0.125	21.05	3231	1.353

**Table 4 micromachines-14-01705-t004:** Heat dissipation data for different layers of fishnet runner at a speed of 1.0.

Layers	Tmax (°C)	Pmax (Pa)	Vmax (m/s)
0	21.06	3033	1.358
1	21.05	3231	1.353
2	21.00	3550	1.358
3	20.95	3849	1.356

## Data Availability

The data presented in this study are available on request from the corresponding author.

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
