# Peer review of "A Novel Swept-Back Fishnet-Embedded Microchannel Topology"

_micromachines, 2023, doi:10.3390/mi14091705_

Round 1

Reviewer 1 Report

1.      As stated in the paper, the heat dissipation is aimed for the single chip of the phased array antennas, so the performance of the chip should be elaborated clearly before the design of the microchannel, and the model should also be explained clearly;

2.      In this paper, only the numerical simulated results are given out, and the performance of the microchannel with topological structure should be simulated comprehensively, and  more detail to the entrie structure should be provided, not just the microchannel.

1.      The quality of English needs improving.

2.      There are many spelling mistakes in the paper, such as μm is written um, and SBFEMCT is written BSFEMCT,  Figure 2 is written Figure 2.2, and so on.

Author Response

Thank you so much for your careful review and constructive suggestions with regard to our manuscript. These comments are very helpful for us to improve our paper. We have revised the manuscript carefully according to all the comments. All the changes have been marked in blue (for modified and added words) in the revised manuscript. The detailed response and revised contents are provided in the attachment.

Reviewer 2 Report

The author has performed interesting study in microchannel. There are some suggestions/comments that needs to be addressed prior to acceptance.

1. The introduction section can be enriched by recent technologies in microchannel heat sink. The following articles can be useful in gaining the prospective: https://doi.org/10.32933/ActaInnovations.45.4 ;   https://doi.org/10.1016/j.est.2023.107548 ; https://doi.org/10.1016/j.aej.2023.02.016; https://doi.org/10.1016/j.ijthermalsci.2020.106609; 10.1016/j.energy.2020.119223; 10.1016/j.ijheatmasstransfer.2017.03.007; 10.1016/j.enconman.2018.07.047; 10.1016/j.rser.2017.04.112;

2. Line 162 and 163 instead of u write down micro symbol.

3.  Why does author consider constant thermophysical properties instead of variable properties. You may have used UDF or polynomial function in ANSYS for properties defination.

4. Dean vortices phenonemena can be represented in better way in Fig.1 Further, the phenomena of bifurcation will be more useful over here compared to this. 

5.  The figures need to be processed like blue background, contour legend value may be written in simple form (not in exponent form), etc. Furhter, the quality needs to be improved. 

6. The dimensions and clear image of all configurations should be shown.

7.  Insert unit in Table 2, 3 and 4.

8.  There is no figure under figure 6 in page 7.

9.   Result and discussion needs to be improved.

Author Response

(The authors gave the same response as above.)

Reviewer 3 Report

The English language is fine and minor editing is required.

Author Response

(The authors gave the same response as above.)

Round 2

Reviewer 1 Report

In order to demonstrate the effectivity of the novel swept-back-fishnet-embedded microchannel, I suggest that the temperature results (simulation or experiment ) of the chip with the traditional microchannel structures could be compared here. 

There are also many spelling mistakes, such as ANSYS workbench and Workbench, which should be consistent, errors in line 274 and 275.

Author Response

(The authors gave the same response as above.)

Reviewer 2 Report

It can be accepted for publication.

Author Response

Thank you very much for your careful review of our manuscript.

Reviewer 3 Report

Thanks for answering the questions and comments.

Author Response

(The authors gave the same response as above.)
